# A transition-based neural framework for Chinese information extraction

**Wenzhi Huang** [1,2], **Junchi Zhang** [2]*, **Donghong Ji** [1]*

**1** Key Laboratory of Aerospace Information Security and Trusted Computing, Ministry of Education, School of Cyber Science and Engineering, Wuhan University, Wuhan, Hubei, China, **2** School of Computer Science and Engineering, Wuhan Institute of Technology, Wuhan, Hubei, China

\* zjc.whu@gmail.com (JZ); dhji@whu.edu.cn (DJ)

**Data Availability Statement:** ACE2005 data used in our paper is owned by Linguistic Data Consortium, The Trustees of the University of Pennsylvania(https://www.ldc.upenn.edu/). Please refer to https://catalog.ldc.upenn.edu/LDC2006T06

## Abstract

Chinese information extraction is traditionally performed in the process of word segmentation, entity recognition, relation extraction and event detection. This pipelined approach suffers from two limitations: 1) It is prone to introduce propagated errors from upstream tasks to subsequent applications; 2) Mutual benefits of cross-task dependencies are hard to be introduced in non-overlapping models. To address these two challenges, we propose a novel transition-based model that jointly performs entity recognition, relation extraction and event detection as a single task. In addition, we incorporate subword-level information into character sequence with the use of a hybrid lattice structure, removing the reliance of external word tokenizers. Results on standard ACE benchmarks show the benefits of the proposed joint model and lattice network, which gives the best result in the literature.

## Introduction

The detection of entity mentions, relations and events are three fundamental tasks in information extraction, which can benefit many downstream tasks such as question answering [1, 2], reading comprehension [3, 4], and stock prediction [5, 6]. Intuitively, these three subtasks are closely correlated in the sense that entity mentions are core components connecting relations and events, while the extraction of relations and events can help improve the accuracy of entity results. For example, consider a Chinese text in Fig 1 which contains three entities "华视(Huashi)", "大厅(hall)", "影迷(fans)" and an event trigger "蜂拥而来(crowd)". If an information extraction model aware of partially identified the event role types *Destination* between "大厅(hall)"影迷(fans)" and *Aartifact* between "影迷(fans)"-"蜂拥而来(crowd)", this result could thus be served as a knowledge source to imply that there probably has a physical location relation (*PHYS*) from "影迷(fans)" to long-range entity "大厅(hall)". In turn, the directed type *PHYS* could improve the confidence of a model to identify that "大厅(hall)" is the *Destination* and "影迷(fans)" is the *Artifact*, instead of the other way around. Therefore, it is beneficial to treat this output structure as a whole without leaving aside strong connections among subtasks.

for the data details. To facilitate following researches, we provide our code and sample data at https://github.com/zjcalva/chinese_information_extraction, which is added in the paper (Section Experiments). e have provide sample dataset (about 3% of full dataset) in this URL. To fully reproduce our study, readers need to replace "test_sample.txt" with full dataset obtained from LDC. The authors confirm that they had no additional access privilege, and that they accessed the LDC data in the same manner described above.

**Funding:** Science Foundation of China (No. 61772378), the Social Science Foundation of Ministry of Education of China (No. 18JZD015), the Natural Science Foundation of Hubei Province (No. 2012FFA088), and the National Key Research and Development Program of China (No. 2017YFC1200500). The funders had no role in study design, data collection and analysis, decision to publish, or preparation of the manuscript.

**Competing interests:** No authors have competing interests.

Previous studies have showed that joint learnings of entities and relations [7–10], entities and events [11–13] can lead to better extraction performance than pipelined methods [14–17]. Due to joint learnings are effective at integrating interactive information between tasks and alleviating the problem of error propagation, there has been work inferencing all subtasks using one single model such as perceptron-based structural predictions [18], contextualized span representations [19], two-channel neural networks [20]. Despite the progress of existing efforts, they still follow a pipelined framework by first predicting entities and event triggers from texts, and then making assignments of relations and arguments to entity-entity pairs and entity-trigger pairs, respectively. To address this issue and perform all subtasks in an integrated graph, we propose a novel transition-based framework [21–23] for Chinese information extraction. The transition-based methods tackle the complex joint search space with a state-transition process, which has been used for structured prediction tasks for NLP, including syntactic parsing [23], entity recognition [24], semantic role labeling [25]. In this paper, we construct the output graph in Fig 1 incrementally from left to right by designing eight transition actions for entity, relation and event recognitions. In this process, all actions can be executed alternately in an interleaving order, which is inspired by the human reading process [26].

Besides, modeling Chinese sentences is challenging compared to languages with explicit word boundaries (e.g. English) [27]. It is a dilemma to decide whether to segment sentences explicitly before involving downstream tasks or directly use the raw characters. The first comes to the consequences of inaccurate boundary cuttings that may bring in noises, whereas the latter is incapable of taking advantage of semantic features of word units, resulting in degradation of the overall performance. To overcome this difficulty, we make use of a lattice architecture

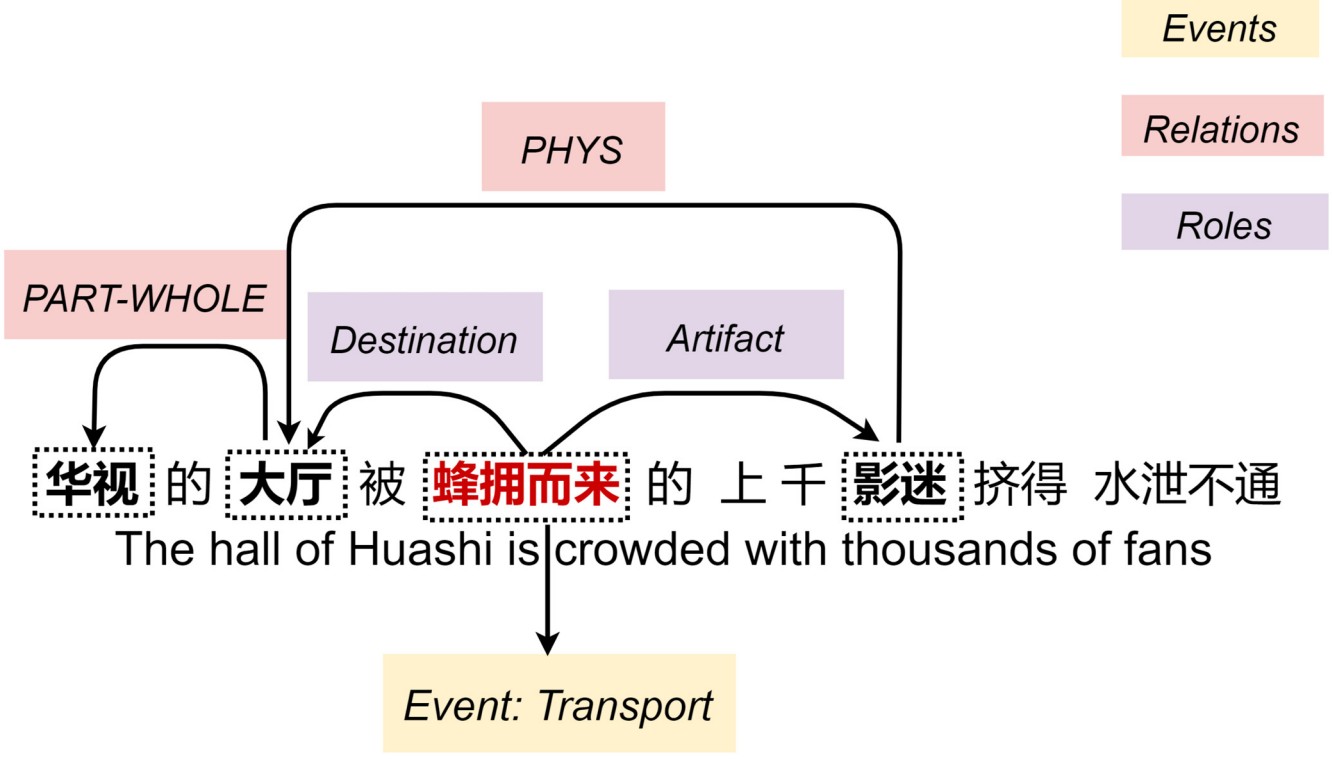

**Fig 1. Example sentence from ACE05 dataset.**

[28] that simultaneously consider both character and word features, leveraging the gate controllers in Long Short Term Memory networks (LSTM) [29] to automatically select the most useful sources for the information extraction subtasks. Different from [28] where off-the-shelf word tokenizers are used, we obtain the most frequent words by using a byte-pair encoding [30] to circumvent segmentation errors.

In addition to the need of encoding words and characters simultaneously, other characteristics of Chinese relation extraction disallow existing transition systems to be applied to our task directly. For example, more than 50% of our Chinese information extraction dataset contains relations or roles between nested named entities and triggers which no existing work deals with. We design a novel transition system that handles such situations.

On standard ACE 2005 benchmarks, our model achieves F1-scores of 81.7%, 50.8%, 68.8% on named entity recognition, relation extraction and event trigger detection, respectively, which are the best in the literature. The main contributions of this paper can be summarized as follows:

1. We propose a novel transition-based system for extracting nested entities, relations, and events in a unified network.

2. We empirically show that lattice LSTM with byte-pair encoding is useful for Chinese information extraction.

3. Experimental results demonstrate that our model significantly outperforms previous methods of using additional knowledge and the state-of-the-art neural models.

## Task definition

According to the Automatic Content Extraction (ACE) 2005 evaluation program, we briefly present the terminologies that relevant to entity relation extraction and event detection. There are four terms involved:

1. **Entity mentions**: An entity is an object or set of objects that belong to a semantic category, such as *Person, Location*. An entity mention is a reference to an entity, usually a noun phrase (NP). We consider the standard PER, ORG, GPE, LOC, FAC, VEH, WEA entity types plus ACE VALUE and TIME expressions as in [11].

2. **Semantic relation**: An particular interests relation that is held between two entity mentions. For example, there is a *PART-WHOLE* relation between "华视(Huashi)" and "大厅(hall)". ACE2005 defines 6 main relation types which are PHYS, PER-SOC, PART-WHOLE, ORG-AFF, ART and GEN-AFF.

3. **Event Trigger**: The keywords that most clearly expresses the occurrence of an event, so that identifying events is equivalent to recognizing trigger words. Event triggers can be verbs, nominalizations, and occasionally adjectives, for instance, the word "蜂拥而来(crowd)" triggers a *Transport* event in Fig 1. Following previous studies [31, 32], we treat 33 event subtypes as separated categories and ignore their hierarchical structure.

4. **Event Argument**: Event arguments are entities that participate as specific roles in the event mention. An argument reflects attributes that an entity carried in an event generally indicates place and time, and specifically have certain values (e.g. CRIME, ARTIFACT). We collapse 8 time-related types into one as in [12], which results in a total of 29 role subtypes.

Formally, given an input sentence represented as a sequence of tokens $C = c_1, c_2, \ldots c_n$, we concentrate on two tasks:

1. **Relation extraction (RE) task** involves predicting a set of entity mentions $E$ and a set of semantic relations $R$ between recognized entity pairs. Note that there may exist nested entities, such as "香港展览中心(Hong Kong exhibition center)" and "香港(Hong Kong)" share common characters prefix. Each relation $r \in R$ can be represented as a tuple $(e_s, e_o, l_{so})$, where $e_s$ and $e_o$ refer to a subject entity and object entity, respectively. $l_{so}$ is a semantic relation category assigned to $e_s$ and $e_o$. An extra None label is added to represent no relation holds for the entity pair.

2. **Event detection (ED) task** involves detecting event trigger sets $T$ and event arguments $A$. In particular, each token $w_i$ will be distinguished to be a true or pseudo trigger mention (i.e. Trigger Identification) and will be further assigned to an event type label $t_i$ if $w_i$ is a positive trigger mention (i.e. Trigger Type Detection). Then, for each trigger $t_i$, participant arguments are assigned to this event trigger by predicting an argument role $a_{ij}$ for all candidate entity mentions $e_j \in E$ in the same sentence (i.e. Argument Role Classification). Depart from most prior work on event extraction, we consider the realistic setting where gold entity labels are not available. Instead, we prepare argument candidates using predicted entity mentions from RE task.

## Methods

### Transition system

The transition-based framework is an effective algorithm that builds structure output incrementally in a step by step process. In particular, a transition system has two key components: (1) transition states and (2) a set of transition actions. In this work, we define each transition state as a tuple $s = (\sigma, \delta, \lambda, e, \beta, E, R)$, where $\sigma$ is a stack storing processed elements (an element can be either an entity or an event trigger), $\delta$ is a stack containing elements that are temporarily popped out of $\sigma$ but will be pushed back later, $e$ is a stack holding words popped off $\beta$, $\lambda$ is a variable storing the current element recognized by $e$, and $\beta$ is a buffer containing unprocessed tokens. $R$ is a set of relations or argument role triples that have been extracted. $E$ is a set of elements that have been recognized. $A$ is a stack storing action histories.

On the other hand, transition actions are used to control how a transition state advance by one step, which should be well designed to ensure that entities, relations and events are extracted in an proper order and covers all possible output graphs. To meet this purpose, we develop eight types of transition actions, including LEFT-*, RIGHT-*, SHIFT, DUAL-SHIFT, DELETE, ELEMENT-SHIFT, ELEMENT-GEN, ELEMENT-BACK. The first four actions are used to extract relations and argument roles, while the last four actions are utilized to recognize named entities. We summarize transition actions as follows:

- LEFT-* pops the top element $e_j$ from $\sigma$ and push it onto $\delta$. It also add a relation or argument role type $l$ between $\lambda(e_i)$ and $\sigma(e_j)$, assuming the directed edge is from $e_i$ to $e_j$.

- RIGHT-* pops the top element $e_j$ from $\sigma$ and push it onto $\delta$. It also add a relation or argument role type $l$ between element $\lambda(e_i)$ and $\sigma(e_j)$ which is similar to LEFT-*, but the direction is opposite from $e_j$ to $e_i$.

- SHIFT pops all the elements from $\delta$ back to $\sigma$ and move element $e_i$ in $\lambda$ to the first element of $\sigma$.

- DUAL-SHIFT is similar to DUAL-SHIFT but additionally copies the element words in $\lambda$ and pushes it onto $\beta$, in order to handle situations where a word is a trigger and also the first word of an entity.

- DELETE simply removes the left-most word off buffer $\beta$.

- ELEMENT-SHIFT pops the front word off $\beta$ and moves it to the front of $e$

- ELEMENT-GEN summarizes all the words in $e$ and generates an entity label or an event trigger label.

- ELEMENT-BACK moves all but the last word from $e$ back to $\beta$. It is designed to tackle overlapping entities and triggers.

To ensure a valid output, each action needs to satisfy certain preconditions. For example, actions that extract relations or argument roles can only be conducted after the action ELEMENT-GEN is performed. Table 1 shows the full list of preconditions.

## Neural transition-based model

The network architecture of our proposed transition method is shown in Fig 2. For a given Chinese text, we first utilize lattice-lstm [28] to encode unigram and bigram characters. Then these hidden representations are fed into a decoder layer to generate a transition state with the guidance of the previous step transition action. At last, all structure features are summarized to predict possible actions for the next transition state.

**Embedding encoder layer.** As presented in Fig 1, for each input Chinese character $c_i$, we represent it as a concatenation of the corresponding character unigram embeddings and character bigram embeddings:

$$\mathbf{x}_i = \mathbf{c}_i \oplus \mathbf{c}_{i,i+1} \qquad (1)$$

where $\oplus$ represents *concatenate* operation. The unigram embeddings are initialized by taking the last layer output of the pre-trained bi-directional language models, termed as BERT [33]. Specially, we use the Chinese version of the base BERT parameters, which is provided by huggingface project(https://huggingface.co/bert-base-chinese).

In addition to character-level features, we also build word-level segments by adopting the Byte Pair Encoding (BPE) [34] algorithm to encode the subword information. The original idea of BPE is to iteratively compress data by merging the most frequent pair of bytes in a sequence as a new byte. Here, we construct the most frequent subwords using Chinese Gigaword corpus (https://catalog.ldc.upenn.edu/LDC2003T09). The subword vector $\mathbf{w}_{b,e}$ that starts from index $b$ to index $e$ is initialized with pre-trained word2vec method [35].

**Table 1. Preconditions of transition actions.**

| Transitions | Preconditions of transition actions |
|---|---|
| LEFT-* | $(\lambda \neq \psi) \wedge (\sigma \neq []) \wedge (j \in T) \wedge (i \in E)$ |
| RIGHT-* | $(\lambda \neq \psi) \wedge (\sigma \neq []) \wedge (j \in E) \wedge (i \in T)$ |
| SHIFT | $(\lambda \neq \psi) \wedge (\sigma = [])$ |
| DUAL-SHIFT | $(\lambda \neq \psi) \wedge (\sigma = []) \wedge (j \in T)$ |
| DELETE | $(\exists j \in \beta) \wedge (e = [])$ |
| ELEMENT-SHIFT | $(\lambda = \psi) \wedge (\exists j \in \beta)$ |
| ELEMENT-GEN | $(\lambda = \psi) \wedge (e \neq []) \wedge (j \notin E)$ |
| ELEMENT-BACK | $(\lambda = \psi) \wedge (e \neq []) \wedge (j \in E)$ |

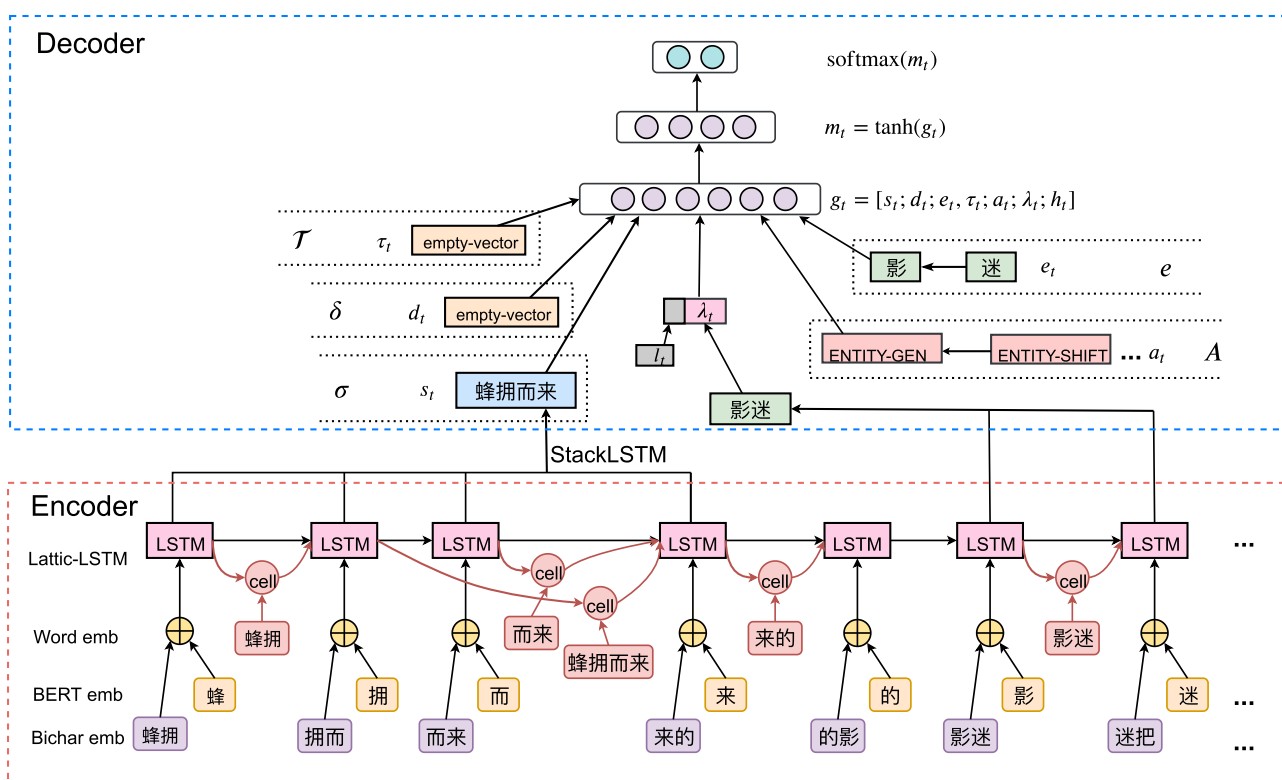

**Fig 2. Our encoder-decoder model for joint entities, relations and events extraction.**

**Lattice LSTM encoder layer.** We represent input sentences with a lattice structure balancing word and character features dynamically with the gate mechanism [28]. At each step, lattice lstm takes a word embedding $\mathbf{w}_{b,e}$ and a character embedding $\mathbf{x}_i$ as input, first calculate three types of word gates and the cell state as follows:

$$
\begin{bmatrix} \mathbf{i}_{b,e} \\ \mathbf{f}_{b,e} \\ \tilde{\boldsymbol{c}}_{b,e} \end{bmatrix} = \begin{bmatrix} \sigma \\ \sigma \\ tanh \end{bmatrix} \left( \mathbf{W}_s^{\top} \begin{bmatrix} \mathbf{w}_{b,e} \\ \vec{\mathbf{h}}_b \end{bmatrix} + \mathbf{b}_s \right)
\tag{2}
$$

$$
\mathbf{c}_{b,e} = \mathbf{f}_{b,e} \odot \mathbf{c}_b^c + \mathbf{i}_{b,e} \odot \tilde{c}_{b,e}
$$

where $\vec{\mathbf{h}}_b$ is hidden vector started at $b$. $\sigma$ is Sigmoid function. $\mathbf{W}_s^{\top}$ and $\mathbf{b}_s$ are model parameters. $\mathbf{c}_{b,e}$ is the memory cell of the shortcut path starting from character $c_b$ to character $c_e$.

An additional gate is used to obtain the cell state of the character $c_i$:

$$
\mathbf{i}_{b,i} = \sigma \left( \mathbf{W}^{g\top} \begin{bmatrix} \mathbf{x}_i \\ \mathbf{c}_{b,i} \end{bmatrix} + \mathbf{b}^g \right)
\tag{3}
$$

Then input gate $\mathbf{i}_{b,i}$ and $\mathbf{i}_i$ should be normalized to ensure their sum equals to $\mathbf{1}$:

$$
\begin{aligned}
\boldsymbol{\alpha}_{b,i} &= \frac{exp(\mathbf{i}_{b,i})}{exp(\mathbf{i}_i) + \sum\limits_{\mathbf{i}_{b',i} \in \mathbb{I}_i} exp(\mathbf{i}_{b',i})} \\[2ex]
\boldsymbol{\alpha}_i &= \frac{exp(\mathbf{i}_i)}{exp(\mathbf{i}_i) + \sum\limits_{\mathbf{i}_{b',i} \in \mathbb{I}_i} exp(\mathbf{i}_{b',i})}
\end{aligned}
\tag{4}
$$

The final forward lattice LSTM representation $\vec{\mathbf{h}}_i$ of character $c_i$ is calculated as:

$$
\begin{aligned}
\begin{bmatrix} \mathbf{o}_i \\ \mathbf{f}_i \\ \tilde{\boldsymbol{c}}_i \end{bmatrix} &= \begin{bmatrix} \sigma \\ \sigma \\ tanh \end{bmatrix} \left( \mathbf{W}^{\top} \begin{bmatrix} \mathbf{x}_i \\ \vec{\mathbf{h}}_{i-1} \end{bmatrix} + \mathbf{b} \right) \\[2ex]
\mathbf{i}_i &= \mathbf{1} - \mathbf{f}_i \\[1ex]
P\mathbf{c}_i &= \sum\limits_{\boldsymbol{c}_{b,i} \in \mathbb{C}_i} \boldsymbol{\alpha}_{b,i} \odot \boldsymbol{c}_{b,i} + \boldsymbol{\alpha}_i \odot \tilde{\boldsymbol{c}}_i \\[1ex]
\vec{\mathbf{h}}_i &= \mathbf{o}_i \odot tanh(\mathbf{c}_i)
\end{aligned}
\tag{5}
$$

where $\mathbf{W}^{\top}$ and $\mathbf{b}$ are the model parameters. Due to Eq 5 has a more complex memory calculation step compared to traditional LSTM [29], which can take account of all the matched input subwords. Therefore, Lattice lstm is more suitable for Chinese texts to integrate character and word semantics simultaneously.

Finally, to obtain a bi-directional representation $\mathbf{h}_i = \vec{\mathbf{h}}_i \oplus \overleftarrow{\mathbf{h}}_i$ for character $c_i$, the backward lattice LSTM vector $\overleftarrow{\mathbf{h}}_i$ is computed using Eqs 2–5 in the reversed order of input sequence.

**Transition state representation.** Given transition State $s = (\sigma, \delta, \lambda, e, \beta, E, R)$, we encode each component as follows:

1. For unprocessd characters $C = c_i, c_2, \ldots c_n$, the buffer $\beta$ is first represented by pushing all lattice LSTM outputs $H = h_1, h_2, \ldots h_n$ onto a vanilla stack the reversed order.

2. For $\sigma$, $\delta$, $e$ and $A$, which involve popping out top elements, we use Stack LSTM [23] for encoding. In the initial state, the stacks $\sigma$, $\delta$, $e$ and $A$ all has an zero-vector indicating an empty-stack. The calculation process of all stacks are similar, we thus take $\sigma$ as an example:

$$
\sigma_i = \text{StackLSTM}[e_1, e_2, \ldots e_m]
\tag{6}
$$

where $e_i$ denotes the representation of the entities recognized at time step $i$. Note that an element $e_i$ can be either an entity or a event trigger.

**Action decoder layer.** For a given sentence, we find the best transition sequence by taking the action with maximum probability at every step. Each action is decided according to the current state encoding. Denote the feature vector for the transition state at $t$ as $g_t$, which

consists of the current $\sigma$, $\delta$, $e$, $A$, $\lambda$, $\beta$, as shown in Fig 2. We have representation of the state as:

$$g_t = [s_t; d_t; e_t; a_t; \lambda_t; h_t] \tag{7}$$

A feed-forward neural network with tanh activation function is adopted to transform feature $g_t$ to action prediction space:

$$m_t = \tanh(\mathbf{W}_m g_t + b_m) \tag{8}$$

$$p(z_t | m_t) = \frac{\exp(\mathbf{u}_{z_t}^\top m_t + b_z)}{\sum_{z\beta \in v(S,A)} \exp(\mathbf{u}_{z\beta}^\top m_t + b_z)} \tag{9}$$

where $\mathbf{W}_m$, $\mathbf{u}_z$ and $b_m$, $b_z$ are trainable parameters. The set $v(S, A)$ represents the set of valid candidate actions. $z_t$ is the current predicted transition action.

For network training, we minimize the negative log-likelihood of the corresponding gold action:

$$L(\theta) = -\frac{1}{T} \sum_t \log p(z_t | m_t; \theta) \tag{10}$$

where $T$ is the size of the gold action sequence.

## Experiments

### Experimental settings

**Dataset.** We perform experiments on the publicly available dataset: ACE 2005 Chinese corpus (https://catalog.ldc.upenn.edu/ldc2006t06). The dataset contains 633 documents, which were collected from Newswires (NW), Broadcast News (BN), Weblog (WL). Due to that there is relatively little study on joint all subtasks for Chinese, we randomly split the ACE 2005 dataset into 8:1:1, with 507 documents for training, 63 documents for validation and 63 documents for testing. In addition, for named entities, we take nested entities that contain over-lapping spans into consideration. For relations and argument roles, we use a "None" type to represent there is no semantic connections hold between entities and event triggers. Our code and sample test results are available (https://github.com/zjcalva/chinese_information_extraction).

**Evaluation metrics.** We use Precision (P), Recall (R) and F-Measure (F1) on both entities, relations and events. An entity is considered correct if we can identify its head and the entity type correctly. A relation instance is regarded as correct when its relation type and the head offsets of two corresponding entities and entity types are both correct. An event trigger is considered correct if both its offset and event subtype is correct. An argument role is regarded as correct when argument offset, role type and the corresponding triggers are both correct.

**Pre-processing.** To represent input Chinese sentences, we use the same character unigram embeddings, bigram embeddings and word embeddings with [36], which pretrain those embeddings using word2vec [35] on Chinese Gigaword corpus. The vocabulary of subword is constructed with 150000 merge operations in byte-pair encoding. To accelerate the subwords construction process of lattice network inputs, trie structure [37] is used. Note that all the embeddings are fine-tuned during training.

**Hyper-parameter settings.** All the hyper-parameters are tuned by selecting the best model with early stopping using the evaluation results on the validation set. Specifically, we set the embedding sizes of character unigram,bigram and subword all to 50 dimensions. Dropout [38] is set to 0.33 on both the character input and the subword input to prevent overfitting.

Adam [39] is applied to optimize the model parameters, with an initial learning rate of 0.015 and a decay rate of 0.05. The batch size and the lattice lstm hidden dimension are set to 32 and 150, respectively.

## Development results

To examine the influence of several key model components, we report entity and relation extraction results on the ACE2005 validation set.

**The influence of lattice LSTM.** Table 2 shows the influence of the encoder layer by comparing the following models:

1. Transition: using a standard LSTM to encode the character embedding of the input sentence, in which BERT embeddings are incorporated.

2. Transition-BERT: BERT embeddings are removed from Transition to see the effect of the pre-trained bidirectional language model.

3. Transition+Bigram: in addition to unigram raw characters, this model also leverages adjacent characters in a sentence to form bigram features, such as "蜂(bee)" can be represented as "蜂拥(swarming)".

4. Transition+Lattice (word): the proposed hybrid model that integrates characters and words in a lattice networkï¼?where Chinese words are obtained by applying Jieba tokenizer.

5. Transition+Lattice (subword): similar to Transition+Lattice (word), but subwords are built by the byte-pair encoding algorithm.

With raw unigram input, baseline **Transition** yeilds 73.1% and 45.3% F1-scores on entities and relations, respectively. The F1-scores are decreased to 71.5% and 42.2% by removing BERT embeddings to the baseline model, indicating the effectiveness of the bi-directional language model pre-trained on a large amount of plain texts. Compared with the only unigram features, **Transition+Bigram** gives slightly higher F1-scores, which are 74.2% on entities and 45.5% on relations, respectively. The results indicate that bigram information will promote the performance of the baseline model, but the increase in F1-score is not significant.

Finally, although we take multiple strategies to combine character features, we can see that lattice lstm network gives the most high performance on both entity and relation extractions. In particular, subword-based lattice model achieves 81.7% F1-scores on entities and 50.8% on relations, respectively. The results reveal two conclusions: 1) Lattice lstm is effective to integrate character and word sequence information in a hybrid framework; 2) Lattice +Subword works better than Lattice+Word indicating the usefulness of allevating segmentation errors. With above observations, we take subword-based lattice lstm model for subsequent experiments.

**Table 2. The influence of lattice LSTM results on the ACE2005 test set.**

| Models | Entity Recognition | | | Relation Classification | | |
|---|---|---|---|---|---|---|
| | P | R | F1 | P | R | F1 |
| Transition | 75.3 | 71.0 | 73.1 | 55.7 | 39.0 | 45.3 |
| Transition—BERT | 74.5 | 68.6 | 71.5 | 44.8 | 39.8 | 42.2 |
| Transition + Bigram | 78.1 | 70.7 | 74.2 | 58.0 | 41.4 | 45.5 |
| Transition + Lattice (Word) | 83.5 | 79.2 | 81.3 | 55.3 | 48.2 | 49.9 |
| Transition + Lattice (Subword) | 84.2 | 79.3 | **81.7** | 56.5 | 48.8 | **50.8** |

**Table 3. Joint model and pipeline model results on the ACE2005 test set.**

| Models | Entity Recognition | | | Relation Classification | | |
|---|---|---|---|---|---|---|
| | **P** | **R** | **F1** | **P** | **R** | **F1** |
| Lattice-transition-pipeline | 81.0 | 78.3 | 79.6 | 44.2 | 51.4 | 47.2 |
| Lattice-transition-joint | 84.2 | 79.3 | **81.7** | 56.5 | 48.8 | **50.8** |

**Joint model vs pipeline model.** To evaluate the advantages of the joint model (Lattice-transition-joint), we construct a pipelined version (Lattice-transition-pipeline) by first performing entity recognition with relation and event-related transition actions removed from our joint model including "LEFT-*", "RIGHT-*", "SHIFT", "DUAL-SHIFT", then modify some actions for detecting relations and events in the sentence independently.

Table 3 lists the results on the ACE2005 test set. The first observation is that the pipelined model is a strong baseline, which gives the F1-score of 79.6% on entity recognition and 47.2% on relation extraction. However, by treating entities, relations, and events as a single task, the joint transition model outperforms the piplined model by 2.1% on entities and 3.6% on relations. We attribute these improvements as twofold: 1) Modeling the dependencies of entities and the semantically related tasks are effective for reducing error propagation. 2) Our joint model is able to capture label correlations between tasks not only at the encoding part but also at the decoding stage, which is, in turn, also beneficial for entity recognition.

## Comparison results

**Baselines** To demonstrate the effectiveness of various kind of models, we implement several advanced approaches as baselines for comparison:

- **Word-Tree-Structure** [40] is a word-based joint learning model, where word segments are obtained by applying Jieba tokenizer (https://github.com/fxsjy/jieba). In particular, a bidirectional LSTM network is leveraged as the shared encoder. We first extract entities and event triggers as two sequence labeling tasks with BILOU output scheme. Then we construct the tree structure of input word sequences with Stanford dependency parsing (https://nlp.stanford.edu/software/stanford-dependencies.shtml) to obtain shortest dependency paths between entities and event triggers. Finally, recognized element type embeddings and bi-lstm vectors along the shortest dependency path are concatenated as input for relation and argument role classifications.

- **Char-BERT-pipeline** [33] learns extractions of entities, relations and events in a pipelined process. It employs pre-trained Chinese BERT-base embeddings as a contextualized character encoder which is fine-tuned during model training. To predict task labels, we simply add a linear transformation layer on top of individual BERT outputs and use softmax function to normalize label vectors. It has been shown that BERT is a powerful representation method, which contains hierarchical lexical, syntactic and semantic knowledges [41]. Hence, we believe it is a strong baseline for comparison.

**Analysis** Table 4 shows the results of different types of approaches. The first observation is that despite the end-to-end learning process of **Word-Tree-Structure**, it performs relatively poor on both entity recognition (72.6%) and relation extraction (39.3%). One possible reason is that it copes with subtasks only on word-level without taking advantage of character-level information. On the other hand, it suffers from noise parsing results incorporated by off-the-shelf word segmenters and dependency parsers, which is prone to affect downstream applications.

**Table 4. Entity and relation results compared to previous systems on the ACE2005 test set.**

| Models | Entity Recognition | | | Relation Classification | | |
|---|---|---|---|---|---|---|
| | P | R | F1 | P | R | F1 |
| Word-Tree-Structure | 79.0 | 67.2 | 72.6 | 43.6 | 35.7 | 39.3 |
| Char-BERT-pipeline | 77.6 | 73.1 | 75.3 | 45.0 | 42.3 | 43.6 |
| Lattice-transition-joint | 84.2 | 79.3 | **81.7** | 56.5 | 48.8 | **50.8** |

Second, **Char-BERT-pipeline** shows a noticeable improvements (2.7% on entities and 4.3% on relations) compared to **Word-Tree-Structure** even with a separated extraction process. This result demonstrates that hierarchical information contained implicitly in the pretrained deep language model is a key factor to attain a feasible performance.

The final observation is that our model **lattice-transition-joint** significantly outperforms the existing methods on both two tasks. In particular, for entity recognition, our model achieves 9.1% improvements compared to [40] and 6.4% improvements compared to BERT-based model [33]. This result indicates the importance to exploit long-distance and cross-task dependencies between entities, relations and events. By allowing entity information propagate through transition states, our model reaches the best 50.8% F1-scores on relation extraction, demonstrating it is beneficial to use the partial identified graph as a source of features for more informed constructions of output structures.

## Event extraction results

In addition to extracting entities and relations, our transition methods also extract event triggers and argument roles. We compare their results with our pipeline method as well as previous work. As it can be observed from the Table 5, by incorporating the relation information generated in state-transition process, **Lattice-transition-joint** can improve the performance than the pipelined method **Lattice-transition-pipeline** by 1.8% and 5.3% on event trigger detection and argument role classification, respectively. This demonstrates that relation attributes truly reinforce the event recognition. And incremental joint learning is capable of delivering the features of relations to the event modeling process. Additionally, our joint model significantly outperforms **Word-Tree-Structure** and **Char-BERT-pipeline** with a performance gain of about 3% F1-scores on event triggers and 7% F1-scores on argument roles. We attribute the failure of previous approaches to error propagation from the upstream task and the incapacity of handing nested entities.

## Related work

Entity recognition [28, 42], relation extraction [14, 43] and event detection [44, 45] are fundamental tasks in NLP, which have drawn much attention in recent years. For English, joint

**Table 5. Event trigger and argument role results on ACE2005 test set.**

| Models | Event Trigger Detection | | | Argument Role Classification | | |
|---|---|---|---|---|---|---|
| | P | R | F1 | P | R | F1 |
| Word-Tree-Structure | 67.7 | 58.3 | 62.6 | 49.2 | 39.8 | 44.5 |
| Char-BERT-pipeline | 62.3 | 68.9 | 65.4 | 48.5 | 51.5 | 50.0 |
| Lattice-transition-pipeline | 62.3 | 72.2 | 66.9 | 52.8 | 52.0 | 52.4 |
| Lattice-transition-joint | 65.5 | 72.5 | **68.8** | 56.2 | 59.2 | **57.7** |

methods include integer linear programming models [46, 47], feature-based structured learning models [7, 48] and neural network models [13, 40, 49]. Methods considering all subtask simultaneously includes contextualized span representations [19], interactive two-channel neural networks [20]. The most closed work to ours is [18], which jointly decodes all subtasks by designing two types of decoding actions in the structured perceptron algorithm. However, they heavily rely on manually designed indicator features struggling to capture sufficient discriminative information.

For Chinese, there have been word-based [50] and character-based [28, 51] models for Chinese named entity recognition. Peng and Dredze [52] propose an integrated model to process multi-task learning that allows for joint training learned representations. By integrating latent word information into character-based LSTM-CRF, Zhang and Yang [28] propose lattice LSTM representation for mixed characters and lexicon words. For relation and event extraction, there have been kernel-based methods [8, 43], feature-based methods [8, 31, 32] and neural network methods [53–55]. All of the models require segmentation first and thus suffer from the potential issue of error propagation. To our knowledge, we are the first to design a neural network to jointly recognize Chinese named entities, relations and events without the need of word segmentation.

## Conclusion

We proposed a transition-based framework that treats all Chinese subtasks of entity recognition, relation extraction and event detection as a single task without leaving aside mutual benefits among tasks. In particular, we designed eight transition actions to ensure all subtasks are identified in a proper order. In addition, to unburden the need of word segmentation before information extraction, we adopt a lattice network to integrate subword features into character-level sequences. Experimental results on the standard ACE2005 benchmark show that our model achieves superior performance over both word-based joint learning method and character-based pipelined BERT network, reaching a state-of-the-art result on Chinese information extraction.

## Acknowledgments

We would like to thank the anonymous reviewers for their many valuable comments and suggestions.

## Author Contributions

**Data curation:** Wenzhi Huang.

**Formal analysis:** Wenzhi Huang.

**Methodology:** Wenzhi Huang.

**Software:** Junchi Zhang.

**Supervision:** Junchi Zhang.

**Validation:** Junchi Zhang.

**Visualization:** Junchi Zhang.

**Writing – original draft:** Wenzhi Huang.

**Writing – review & editing:** Donghong Ji.

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
