## [Decision Letter · Decision Letter 0]

21 May 2020

PONE-D-20-12576

A Transition-based Neural Framework for Chinese Information Extraction

PLOS ONE

Dear Dr. Huang,

Thank you for submitting your manuscript to PLOS ONE. After careful consideration, we feel that it has merit but does not fully meet PLOS ONE’s publication criteria as it currently stands. Therefore, we invite you to submit a revised version of the manuscript that addresses the points raised during the review process.

Please revise the paper by considering the reviewer's comments.

We look forward to receiving your revised manuscript.

Kind regards,

Jie Zhang

Academic Editor

PLOS ONE

Journal Requirements:

3. Please ensure that the manuscript contains adequate English translations.

6. Please include a copy of Table 6 which you refer to in your text on page 9.

7. We note you have included a table to which you do not refer in the text of your manuscript. Please ensure that you refer to Table 3 in your text; if accepted, production will need this reference to link the reader to the Table.

Reviewers' comments:

Reviewer's Responses to Questions

**Comments to the Author**

1. Is the manuscript technically sound, and do the data support the conclusions?

Reviewer #1: Partly

2. Has the statistical analysis been performed appropriately and rigorously? 

Reviewer #1: Yes

3. Have the authors made all data underlying the findings in their manuscript fully available?

Reviewer #1: No

4. Is the manuscript presented in an intelligible fashion and written in standard English?

Reviewer #1: Yes

5. Review Comments to the Author

Reviewer #1: Summary: This paper proposes a joint extraction approach for Chinese information. The main idea is to combine lattice LSTM and transition system to build an end-to-end learning network to complete the joint extraction of entities, relations and events. The authors demonstrate improvements over the state of the art on ACE 2005 Chinese datasets.

The paper is thorough in its presentation of the approach and evaluation, and is generally well written and readable. However, it does requires further editing to address numerous grammatical errors, especially with regard to the use of articles.

Although datasets cannot be publicly shared, authors can provide an online replication package that includes some test samples, allowing reviewers and readers to validate the analyses and data filtering that was performed.

6. PLOS authors have the option to publish the peer review history of their article (what does this mean?). If published, this will include your full peer review and any attached files.

Reviewer #1: No

---

## [Author Response · Author response to Decision Letter 0]

19 Jun 2020

Reviewers' comments:

5. Review Comments to the Author

Reviewer #1: Summary: This paper proposes a joint extraction approach for Chinese information. The main idea is to combine lattice LSTM and transition system to build an end-to-end learning network to complete the joint extraction of entities, relations and events. The authors demonstrate improvements over the state of the art on ACE 2005 Chinese datasets.

The paper is thorough in its presentation of the approach and evaluation, and is generally well written and readable. However, it does requires further editing to address numerous grammatical errors, especially with regard to the use of articles.

Although datasets cannot be publicly shared, authors can provide an online replication package that includes some test samples, allowing reviewers and readers to validate the analyses and data filtering that was performed.

---Response: Thank you for your comments, grammatical errors are addressed, test samples and replication package are available at https://github.com/zjcalva/chinese_information_extraction, which is added in the paper(Section Experiments)

---

## [Editor Report · Decision Letter 1]

23 Jun 2020

A Transition-based Neural Framework for Chinese Information Extraction

PONE-D-20-12576R1

Dear Dr. Huang,

We’re pleased to inform you that your manuscript has been judged scientifically suitable for publication and will be formally accepted for publication once it meets all outstanding technical requirements.

Kind regards,

Jie Zhang

Academic Editor

PLOS ONE
---

## [Editor Report · Acceptance letter]

6 Jul 2020

PONE-D-20-12576R1 

A Transition-based Neural Framework for Chinese Information Extraction 

Dear Dr. Huang:

I'm pleased to inform you that your manuscript has been deemed suitable for publication in PLOS ONE. Congratulations! Your manuscript is now with our production department. 

Kind regards, 

on behalf of

Dr. Jie Zhang 

Academic Editor

PLOS ONE